# Recent Advances in Woven Spacer Fabric Sandwich Composite Panels: A Review

**DOI:** 10.3390/polym14173537

**Published:** 2022-08-29

**Authors:** Yeran Wang, Junmei Liu, Lixia Jia, Zhenhong Chen

**Affiliations:** 1College of Textile and Garments, Hebei University of Science and Technology, Shijiazhuang 050018, China; 2Hebei Textile and Garment Technology Innovation Center, Shijiazhuang 050018, China

**Keywords:** core yarn structure, cross-linking structure, filling process, molding process, sandwich structure, mechanical property

## Abstract

Because of the advantageous characteristics of strong integrity, lightweight, high performance, and various designs, woven spacer fabric (WSF) and its composite are extensively used in construction, traffic, and aerospace, among other fields. This paper first describes the WSF structure, including core yarns and cross-linking, and then discusses the influence of the processing parameters, among angle of the wall decisive the failure mode on the plate properties. Moreover, we summarize the molding and filling technology of WSF composite sandwich panels and discuss the process order, resulting in a significant effect on the stiffness of the sandwich composite plate; the current processing is mostly hand lay-up technology. In addition, we introduce the core and matrix material of the sandwich composite plate, which are mainly polyurethane (PU) foam and epoxy resin (70% of matrix material), respectively. Finally, the mechanical properties of WSF composite sandwich panels are summarized, including bending, compression, impact, shear, and peel properties. Factors influencing the mechanical properties are analyzed to provide a theoretical basis for future plate design and preparation.

## 1. Introduction

A sandwich composite plate is made up of reinforcing textiles and core materials, which are lightweight and have high production efficiency, excellent performance, and energy-conservation properties. Consequently, it has been widely used in many fields, including civil and mechanical engineering and biological engineering, as well as in aerospace planes, wallboards, ships, cars, and cold chain cars. Depending on the reinforced structure (two-dimensional (2D) and three-dimensional (3D)), the carrying capacity of the sandwich composite plates varies. The traditional sandwich composite plates are made up of 2D upper and lower surface layers and a core layer, which is prone to the debonding problem [1] in the loading, leading to the failure of the plate. Therefore, Kamble et al. [2], Neje et al. [3], Drake et al. [4], and Che et al. [5] studied the suture structure. In a study by Che et al. [5], the suture structures were found to have higher shear stiffness and extra-facial compressive strength than the traditional sandwich structures. However, compared with the cross-linking structure of the WSF, the joint point strength of the suture structure is relatively low and prone to failure [2]. Additionally, the WSF sandwich composite plate has very high cementation strength [2] and skin-core layered resistance [6], which solve the stratification problem of traditional sandwich composite materials. It has potential applications in heat/sound insulation performance [7,8], construction [9], electronic textiles [10], shock absorbers [11], biomedical engineering [12], fan blades, vehicles [13], and so on.

For the structure and weaving of WSF, Geerinck et al. [11], Manjunath et al. [14], and Neje et al. [15] tested the bending and compression properties of woven structures, such as “T” type, “8” type, rectangle, “X” type, and hexagon. They showed that the mechanical properties of the composites are determined by the geometry and structure of the WSF. With the preparation system constantly improving, Geerinck et al. used a standard VSi42 loom with 24 heddle frames for 3D Weaving [11], moving from manual weaving to automatic industrial processing, which promoted the diversity preparation of WSF.

WSF has received a lot of attention in various fields because of its superior integrity and stable structure [16]. In recent years, the preparation methods and mechanical properties of WSF sandwich composite plate have been extensively investigated. From fiber selection to composite plate preparation, the process of producing the WSF sandwich composite plate is long, and there are many factors affecting the final performance of the plate, so most researchers only analyzed the macroscopic influence of a single factor for a performance index of WSF sandwich composite plate, such as reinforcing fiber material, core material, the structure design and weaving of the WSF, processing technology, and mechanical-properties testing, as shown in Table 1. In order to systematically analyze the influence of the mechanical properties of the WSF sandwich composite plate, the reinforcement structure, processing technology, and mechanical tests of WSF sandwich composite plate are reviewed and analyzed in this paper. This paper will provide new research directions and ideas for the research and application of WSF sandwich composite plates and also lay a certain theoretical foundation for the microscopic analysis of the loading failure mode and damage mechanism of the plate by our subject group.

## 2. Research Progress of WSF

### 2.1. Material

The materials used in WSF are generally high-performance fibers (such as polyester fiber, aramid, and polypropylene) and inorganic fibers (such as glass fiber and basalt). Among carbon fibers, glass fiber and basalt fiber have good mechanical properties and a higher modulus. Compared with these fibers, carbon fiber is more prone to brittleness fracture and integrity damage than glass fiber, but basalt fiber is less prone to fractures [17]. Polyester fiber, aramid, and polypropylene have higher strength and toughness and better integrity than glass fiber [11]. To optimize the performance of a single fiber, many researchers tried blending high-performance fibers with inorganic fibers to form mixed yarn, producing a new arrangement in fabric geometry that will affect the mechanical properties of composites [27,28,29]. For example, Zhong et al. [30] studied the core spun yarn produced by polypropylene fiber and basalt filament and found that the breaking strength of wrapped yarn gradually increased with an increase in twist levels within a certain range.

### 2.2. Structure of WSF

At present, the structural design of WSF can be divided into two categories. The first one is the ground fabrics structure design, such as plain and twill. The second WSF design is the internal connection design, including core yarn and cross-linking structure, which is woven through a manual stacking process [8,25,31,32], as shown in Figure 1. The core yarn structure includes an internal design “8” shape [7] (Figure 1a), “S” type, “1” shape, etc. Meanwhile, cross-linking structures, such as internal design, are rectangular (single or double walls (Figure 1f)), triangular (sawtooth) type (Figure 1b), trapezoidal type [3] (Figure 1c), X type (Figure 1d), hexagonal shape (Figure 1e), I type, U type [27], H type [14], etc.

#### 2.2.1. Ground Fabrics Structure Design

The WSF for upper and lower ground fabrics is generally woven through plain, twill, and satin. The plain fabric has the characteristics of many interwoven points and a stable structure, whereas twill and satin fabrics have fewer interwoven points and longer floating lines. Compared with satin and twill structures, the plain fabric structure is tighter, not easy to slip off yarn, has greater stability and symmetry, and has good mechanical properties when woven in the same density [34]. By comparing the influence of different ground-fabrics structures on WSF sandwich composite plates, our research group found that complex ground fabrics have higher bending strength than plain fabric. This conclusion is consistent with a study of hollow WSF sandwich composite plates by Zhang et al. [7], who illustrated that the core material has little effect on bending. It has been shown that the complex ground fabrics–WSF sandwich composite plate can withstand more stress than plain fabric during loading, leading to a higher carrying capacity and higher energy absorption capacity.

#### 2.2.2. Internal Connection Design of WSF

(1)Core yarn structure design

WSF consists of upper and lower layers connected with core yarn, which divides them into weft and warp junctions. For the weft junction, the woven field was only recently studied, but no test analysis has been performed on its mechanical properties because of the complex process and poor design flexibility. The warp junction is more extensive and is woven by controlling the feeding of warp yarn (ground warp and junction warp) under uniform tension and at the proper rate, using a 3D weaving loom, sample weaving machine, rapier loom, and so on [8]. The warp junction yarn has the “8” type (double junction) and “S” type (single junction) structures. The anisotropy of the “S” structure [9,35] is much greater than the “8” structure [32,36], producing an “8” structure with higher stress concentration, disposing it to damage. In contrast, in the “S” structure, stress is more dispersed, which increases the stiffness in loading. In addition, the high compliance and instability along the warp direction could enhance stiffness [37]. Therefore, a weft direction plate has a better bending performance than a warp direction plate [38]. The process parameters that were set for the “8” and “S” structures in previous studies are shown in Table 2.

An analysis of the WSF sandwich composite plate shows that the core yarn height, core yarn density, and thickness of ground fabrics have a certain effect on its performance. Increasing the core yarn density is associated with more overlap and coordination, thus bringing a higher bearing capacity [21]. According to Figure 2e, due to the special characterization mode of multiple fractures of the pile yarns, the strength was increased during compression [8]. For low-core yarn with a high distribution density to achieve improved stiffness, the results revealed that it has good compression resistance, shear strength, and higher unit height energy absorption. Moreover, when the pile yarn height increases, the Young’s modulus and shear modulus are reduced, thus decreasing plate stiffness [35]. In addition, increasing the ground fabrics’ thickness affects the mechanical properties and tensile failure resistance of the WSF sandwich composite plate [39]. Li et al. [40] showed that the ground fabrics thickness can significantly improve the anti-bending load capacity of the WSF sandwich composite. Notably, in addition to the core yarn structural design, Kim et al. [41] studied a way of weaving Truss woven Sandwich by using yarn interleaved between the two panels (Figure 2f I), demonstrating that its strength and elastic modulus were much higher than that of the “8” type structure. This is because the Truss woven Sandwich could provide excellent resistance of WSF sandwich composite, which broke perpendicular to the longitudinal direction, but hardly broke at the intersections adhesively bonded to the neighboring struts during loading (Figure 2f II). Thus, the Truss woven Sandwich achieved improved compression resistance and shear strength performance.

The 3D double-layer WSF uses an interlayer design in its weaving that needs to solve the inclination directions in the two adjacent spaces by applying forces opposite to the external and internal fabric in the weft direction [31]. Chinese researchers found that the core height of a double-layer structure affected the performance of the plate, with a height of 4 cm + 8 cm producing higher stress core yarn of 6 cm + 6 cm [42]. In addition, a double-layer structure had a better energy absorption capacity and stronger bearing capacity than a single-layer structure. Compared with bonded double-layer WSF, the integral double-layer WSF has higher specific stiffness and compressive strength [31].

**Table 2 polymers-14-03537-t002:** Process parameters of core yarn–WSF.

Structure	Height (mm)	Core Yarn Density (ends/cm)	Ground Fabrics Thickness (mm)	Type of Plate	Mechanical Property (MPa)	Reference
“8” type	4,6,8,10	P: 8.14–8.24	-	H	FC:1.23–8.16 EC:20.52–33.50 (P); 332.84–65.05 (F) S:0.58–2.91 (P); 0.76–6.03 (F).	[35]
	5,10,15,20,25	1.58–2.08,	0.46–0.75	H	-	[43,44,45]
	30	1.33–1.54	0.5	F; H	B:34.9–69.5 (Hollow); 89.6 (filling)	[23]
	1.94–2.15	-	0.65–1.59	H	B: 233.64–326.55 (P) 239.38–363.09 (F) FC: 109.42–199.01(P) 164.79–214.85(F)	[12]
	-	21	-	H	B: 20–60; FC: 4.2–5.1	[7]
	3,5,8,10,12	-	0.36,0.58	H	B: 14–144; T: 12.98–44.18; EC: 8.99–32.08.	[39]
	3,6,10	P: 2.08–2.31 F: 23.26–26.32	0.44–0.58	H	-	[46]
	-	0.45–2.08	-	F	B: 1–2.5 (P)	[19]
	30	-	0.6	H	B: 3.73	[33]
	20.8	-	0.6	F; H	-	[24]
	4.06,4.08	2.14,2.16	0.38–0.92	H	-	[40]
	10.35,9.62,9.68	2,3,6	0.91,1.01,0.77	H	FC:3.9–24.6	[8]
“S” type	10	P: 8.25	-	H	FC:3.21 EC: 21.89 (P); 59.26 (F) S: 0.81 (P); 1.27 (F)	[35]

“H” is a hollow WSF sandwich composite plate, “F” is a filling WSF sandwich composite plate. “P” is Warp, and “F” is Weft. “FC” is flatwise compression, and “EC” is edgewise compression. “B” is bending, “S” is shear, and “T” is tensile.

(2)Cross-linking Structures’ Design

The WSF of internal fabric connection belong to cross-linking structures, and each fabric part of its cross-linked structure is a 2D fabric that is woven with a pass sword loom [14,15,28], 3D loom [11], and other equipment. Its length can be controlled by changing its weft insertion [47], whereas the thickness and weaving pattern can be controlled by changing the weaving mode [11]. Furthermore, the warp yarn is divided into two groups, one is weaving upper and lower ground fabrics, and the other is weaving the middle layer. For aside inserts, the wall thickness of the cross-linking structures–WSF differs from the surface thickness due to the weaving route of the hexagonal WSF (shown in Figure 1e), resulting in a thicker middle layer compared with other parts [15]. The common structural shapes of cross-linking structure–WFS include rectangular, triangular, trapezoidal, X-shaped, and hexagonal. There process parameters are shown in Table 3. Among them, height and angle are jointly designed to prove the influence of fiber volume content, as shown in Figure 2c [15]. Moreover, for wall thickness, the stress or bearing capacity of bilayer wall structures is greater than that of single-layer walls [33].

A comparison of the properties of rectangular, triangular, and trapezoidal WSF showed that the overall mechanical properties of rectangular WSF are greater than those of triangular and trapezoidal WSF but smaller than those of the “8” core pile structure [33]. This may be mainly because of the angle at which the load-bearing walls were aligned along the loading axis, resulting in an increase in bearing capacity with increasing separation force on the axis (Figure 2d) [3]. A comparison of the performance of single-layer and bilayer structures demonstrated that the performance of bilayer structures was poor. Ghanshyam et al. [6] found that bilayer structures had lower stress but higher bending resistance than single-layer structures (Figure 2d). Additionally, for a multilayer WSF, an investigation of its tensile property on the warp and weft direction found that the number of layers had a significant effect on stress for the warp but did not considerably influence the weft (Figure 2b) [11].

**Table 3 polymers-14-03537-t003:** Process parameters of cross-linking structures–WFS.

Layers	Structure	Height (mm)	Angle (°)	Width (mm)	Ground Fabrics Thickness (mm)	Type of Plate	Mechanical Property (MPa)	Reference
single	rectangular	30,40,51	90	43,53,64,38,28	-	H	-	[15]
		29	90	50	0.9 (SW),0.9 or 0.6 (DW)	H	B:17.09 (SW), 41.64 (BW)	[33]
		21,29,36.5,47.5	90	33,44.5,50,63	0.9	H	B:20–50	[47]
		32,29	90	34,50	0.6			[2]
	triangle	26,28	35,42.5,47	-	0.6	H	B:7–15	[47]
		26	47	-	0.6 or 0.9	H	B:14.51	[33]
	trapezoidal	36,30	70,53,67	38,33	-	H	-	[15]
		27.5	50.5		0.6 or 0.9	H	B: 10.72	[33]
		28,27.5,29,28.5	45,50.5,57,65	30.5,28	0.6	H	B: 8–18	[47]
						H	B: 8–19	
	X-type	35	-	66	-	H	FC: 0.5–0.7	[14]
bilayer	trapezoidal	56.8	45	30	0.9			[2]
		54.5,52,57.5,53.5	45,50.5,57,65	30.5,28	0.6	H	B: 7–16	[6]
	hexagon	56.8	60	30	0.9	F; H	B: 6 (H),8 (F) FC: 40 (H),55 (F)	[11]
		13.2,22,30.8,39.59	60	-	7.62,12.7,17.78,22.86	H	-	[2]

“H” is a hollow WSF sandwich composite plate, “F” is a filling WSF sandwich composite plate, “FC” is flatwise compression, and “B” is bending. “SW” is the single wall, and “BW” is the bilayer wall.

## 3. Processing Technology for WSF Sandwich Composite Plate

At present, the dominant processing technology for the WSF sandwich composite plate includes preparing a composite of resin and filling of foam. This process begins with composite preparation, followed by filling, or filling first and then preparing the composite. The main difference is that the core layer in the first method is impregnated with resin, which can effectively improve the bearing capacity in the loading process, but the internal connection structure combined with the interface with foam is poor, which worsens the integrity of the plate, resulting in shear failure [22,24,44,48]. The second preparation process can produce excellent integrity and good synergy between the filling material and the internal connection structure, but the carrying capacity is worse compared with that of the first preparation method [18,19]. In the actual preparation process, fiber stiffness depends on the choice of the preparation method. For fiber with greater stiffness (such as carbon fiber [12], glass fiber [22], and basalt fiber), the method of first preparing the composite and then filling is generally adopted. In contrast, for fibers with high flexibility (such as polyester fiber [19], aramid, and polypropylene), the method of filling first and then preparing the composite is adopted. This paper summarizes the latest research on the filling process and composite process. The filling and molding process commonly rely on hand lay-up technology, but because of its lower automation and industrialization, it cannot guarantee the uniformity and stability of plates. In particular, the process of filling hollow WSF sandwich composite panels needs to be further improved.

### 3.1. Filling Process for WSF Sandwich Composite Plate

#### 3.1.1. Core Material

Polyurethane (PU) foam, phenolic foam, concrete, and cementation materials are widely used as core materials, but their choice depends on the application field for the WSF sandwich composite plate. Because of its lightweight, high strength, and low thermal conductivity, PU foam composites have been reported to be useful energy-efficient materials in construction, traffic, and cold-chain transportation.

To enhance the mechanical properties of PU foam, Hamid et al. [22] added natural nanostructured zeolite particles to the polyurethane foam, and this greatly enhanced the PU, as shown in Figure 3a. Concrete foam is ordinarily applied in infrastructure construction because, compared with conventional concrete, it is lightweight and more economical; however, this results in low infrastructural stability and strength [49]. Comparing PU foam with concrete foam, Wang et al. [48] found that the performance and energy absorption capacity of PU was generally better than those of concrete foam. However, the fire grade of PU foam is lower. In subsequent studies, Wang et al. [9] used a mixture of cemented foam and mortar to fabricate novel ductile cementitious composites with low thermal conductivity and high fire grade in building fire prevention. In addition, the phenolic foam is exceedingly flame retardant (refractory, low toxicity, and low smoke), exhibits heat resistance and heat preservation, and its bubble hole size reduces while its rigidity increases with increasing foam density. The brittleness, high powder rate, and poor impact resistance properties of phenolic foam is offset by the toughness provided by fiber or polyamide resin. If the polyamide resin toughens the phenolic foam, the compression strength is reduced, but the adopted fiber is different patterns [50] due to the agglomeration of fibers, as shown in Figure 3b.

#### 3.1.2. Filling Technology

The filling technology for the WSF sandwich composite plate is mostly hand lay-up technology. This technique is composed of two methods: filling-the-foaming-agent method and setting-the-temperature-for-the-foaming method. In the foaming-agent method, the fabric is placed in the mold (using molds Figure 4b or not Figure 4a) first, and then the volume expansion of the liquid during the foaming process pushes the two surfaces apart [26,51]. Although this method allows for control of the foaming speed and uniformity of cell size and shape, the design flexibility of the plates is poor, and it is not easy to achieve a large-scale and optimal production level. In addition, few studies report the vacuum-assisted molding process. Vaidya et al. [24] used a 25.4 mm–diameter infusion tube to quickly inject the foam, and it properly controlled the amount of foam injected. However, like temperature foaming, such as epoxy foams, this process is rigorously controlled by using foaming temperature [52] to achieve the desired cell size and shape, uniformity, and improved foam density [53]. In addition, other process parameters such as pressure and concentration of components could also be applied.

### 3.2. Molding Process of WSF Sandwich Composite Plate

#### 3.2.1. Resin Matrix

In the preparation of the WSF sandwich composite plate, epoxy resin, polyester resin, and phenolic resin are generally used, but polyester resin is preferred because of its low price and ease of handling [54,55,56]. Compared with polyester resin, epoxy resin is a widely used polymer, accounting for about 70% of the entire thermosetting resins market, because of its excellent thermal stability, fairly low thermal conductivity, outstanding electrical insulation, and chemical resistance [57,58,59]. The results demonstrated that epoxy-based composites had a stronger bending modulus, core shear stress, and bending stress, but appeared more brittle (Figure 5a) [60,61]. In comparison, polyester resin was prone to deformation and fiber/matrix-interface-layer separation. The brittle and weak shear resistance of the core layer has a significant effect on the properties of the WSF sandwich composite plate. Therefore, in the molding process, epoxy resin is customarily used as a composite material. Phenolic resins are a less-used resin system, except in the high-temperature field and anti-corrosion engineering because of their characteristics of high acid resistance, high-temperature resistance, dimensional stability, etc. [62,63]. However, it is hard to prepare, resulting in WSF sandwich composite plates with poor mechanical strength and toughness. To improve phenolic resin toughness, Ferhat et al. [64] used multi-walled carbon nanotubes and nano-SiO_2_ nanofillers (Figure 5b) within a resin, which improved toughness by 30%.

#### 3.2.2. Molding Technology

The molding technology of WSF includes the hand lay-up process, vacuum infusion process (VIP), RTM process, and slot-coating technology. The hand lay-up process uses two methods to complete the processing: spraying and impregnation. Low-viscosity epoxy resin was sprayed on fabric, and then it was placed in a vacuum chamber and degassed and gassed three times to enhance the infiltration of the resin and remove any bubbles in the wet assemblies [41]. In the process of impregnation, the resin was applied to the ground fabric by using a spray gun, which was convenient for full permeation of both the piles and ground fabric [35] (Figure 6a). It is a method widely applied in the molding process of WSF and WSF reinforcement core (it is a filling, but not complete molding, process). However, hand lay-up process has some limitations, including poor adhesion with fabric, easy penetration in the internal structure of the fabric, and uneven coating thickness. Consequently, the slot-coating technology was proposed by Doyen et al. [66] for use as a positive pump to control the thickness of the coating layer on upper and lower ground fabric through simultaneous coating (Figure 6b) [66] during the WSF molding process.

The VIP is divided into vacuum conditions and semi-vacuum conditions. In vacuum conditions (Figure 6c I), VIP is currently only applied in the molding process of hollow WSF sandwich composite plate, and not in the process for the WSF reinforcement core, which is difficult because incomplete impregnation easily emerges. Then, in the molding process of WSF, the core bar inserted into the hollow of the fabric is wrapped with non-porous Teflon sheets after the impregnating and curing of WSF [47]. This process requires a core bar to maintain its 3D shape and prevent damage to the internal structure. Although this method has more advantages, including lower fiber stiffness, it is limited by the characteristics of poor design flexibility and high specification and size of the core bar. For semi-vacuum conditions (Figure 6c II), WSF was impregnated with resin before the vacuum bag was removed to relax the piles to obtain a 3D hollow WSF sandwich composite plate, which potentially has uneven resin thickness and instability on the upper and lower ground plates. Thus, in the curing process, Ghanshyam et al. [33] used a metal frame to hold the 3D structure and avoid instability. Wang et al. [8] combined the hand lay-up process and the VIP. The method is a novelty, which involved coating of the WSF with resin, using the hand lay-up process for the pre-curing step, and then uniformly infusing resin to the fabrication plate by using VIP.

The RTM process involves infusing low-viscosity resin to a vacuum mold to a produce a composite [20]. Compared with the hand lay-up process and the VIP, the RTM is a less studied method, because it requires very high precision for the mold. Low precision will affect the uniformity of the resin distribution for preform when the vacuum is not enough or bubbles exist in the resin fluid, which produces heavy and hard-to-operate plates with limited design flexibility [67]. However, composite plates produced through the RTM process have excellent performance compared to those produced by using the hand lay-up process [20]. The RTM process is inclined to form an interphase between the resin matrix and reinforcement in the vacuum-pressure-improving property [68], but this interphase is not obvious in atmospheric pressure for the hand lay-up process.

## 4. Mechanical Properties of WSF Sandwich Composite Plate

The WSF sandwich composite plate is mainly used in carriages, construction materials, shock absorbers, wind-turbine blades, etc., which require WSF sandwich composite plates with bending, compression, impact, shear, and peel performance. These properties are reviewed in this section. Future mechanical research on WSF sandwich composite panels should use non-destructive testing technology.

### 4.1. Bending Property

In regard to bending performance, the core pile structure–WSF sandwich composite plate is generally reported, including its core material, curvature, additional panels, core pile density, core pile height, and fiber material. The fiber material is diverse in regard to stiffness, modulus, and toughness. Consequently, the fiber-reinforced material studied, similar to glass fiber and carbon fiber, is inclined to breakdown and shear failure, but polyester fiber is easy to stratify. A combination of glass fiber and polyester fiber was studied by Jia [17] in which polyester fiber and woven glass fiber were used for ground fabric and core yarn, respectively. These methods improved core yarn stiffness, kept back the toughness of the ground layer, and improved the properties of the WSF sandwich composite plate. Moreover, the core pile height and density were considered. The regression analysis revealed an optimal bending strength in 1.99 ends/cm^2^ for polyester fiber, and an increase in the core pile height decreased its bending strength, ultimately increasing its bearing capacity [19]. The property of the core material is the difference between PU foam and industrial core materials (such as concrete foam): the PU foam is flexible [22], but industrial materials cause shear fracture failure, which is more obvious with the narrowing of span [9]. A test diagram of the minimum span is shown in Figure 7c. Tohid et al. [23] tested the effect of different curvatures and additional panels on bending performance. The results reported in this study showed that the bending property is optimal when the curvature is 0.014 cm^−1^, and an additional panel has a more significant effect on layers, as shown in Figure 7b.

At the present stage, the cross-linking structures–WSF sandwich composite plate has attracted less research attention than the cross-linking structures–hollow WSF sandwich composite plate. Specifically, the effects of ground-fabric structure, cell density, and height were examined for the cross-linking structures–hollow WSF sandwich composite plate. In the ground fabric structure, the crimped and non-crimped (Figure 7d) were researched. The results revealed that non-crimped structures yield a higher flexural resistance than crimped because of the stretched thread arrangement in the fabric structure, both in the warp and weft directions [29]. In addition, the cell wall is a major load-bearing unit [47]. The cell wall of a rectangular structure–hollow WSF sandwich composite plate has better optimal stress than trapezoidal and triangular ones, and when the space between the cell wall and height decreases, the stress increases [33,47]. A study of four types of structures (Figure 7a) by Manjunath et al. [14] found that the H-type structure’s stress was higher than that of the rectangular type.

### 4.2. Compression Property

This section discusses core pile structure and cross-linking structure–WSF sandwich composite plates and the effect of compression properties. The specific properties that have been studied include flatwise compression, edgewise compression, and dynamic compressive for core pile structure–WSF sandwich composite plate and flatwise compression and edgewise compression for cross-linking structure–WSF sandwich composite plates.

(1)Core pile structure–WSF sandwich composite plate

In regard to flatwise compression, the core pile height, core density, and additional panels have been investigated in the literature. For a higher core pile height, the composite plate is inclined to instability, resulting in decreased compression strength [25]. As the stiffness with core density increases, the flatwise compression strength is also increased. It is also strengthened by toughening the core material or additional panels. Chinese researchers used resin and fiber to toughen the core material, concluding that the composite plate had higher compression properties. In additional panels, Hosur et al. [26] investigated glass fiber and carbon fiber plates to strengthen their dynamic compression, establishing that glass fiber plate has higher stress than carbon fiber plate, and core and integrated core pile provide a synergistic effect, but they are prone to shear deformation, causing delamination between additional panels and core pile structure–WSF sandwich composite plate (Figure 8b).

In regard to edgewise compression, a new type of core material was proposed by Wang et al. [9] and Hamid et al. [22]. Wang et al. [9] tested the cementitious-reinforced WSF sandwich composite plate and found that the peak load was sustained by the core, though multiple shear failures occurred. Compared with edgewise compression, flatwise compression had excellent properties because the enhancement plate and the core material improved its strength. In contrast, Hamid et al. [22] investigated foam-core-reinforced WSF sandwich composite plate and found that the composite had higher energy absorption because of the existence of the foam core, and it had a superior peak load compared to flatwise compression. The reason for these two different conclusions could be the difference in core material. Moreover, our research group discussed the impact property before and after, and the results indicate that, compared with the impact before, the impact of the WSF sandwich composite plate after was extremely prone to stress concentration rather than disperse, thus increasing the risk of integrity failure.

(2)Cross-linking structure–WSF sandwich composite plates

For both flatwise compression and edgewise compression, only the hexagonal WSF sandwich composite plates were investigated, which are elastic and have a cushion structure. In edgewise compression, Ruben et al. [11] investigated the effects of aromatic fiber, polyester fiber, and glass fiber on WSF sandwich composite plates. They found that aromatic fiber and polyester fiber–WSF sandwich composite plates contributed to the ductile failure and wrinkling of walls, thereby improved the edgewise compression strength. For the glass fiber, the failure pattern was the ductile buckling and the classic brittle tearing/breaking of the wall which decreased the compression strength (Figure 8a). For flatwise compression, Zhu et al. [69] examined the compression behavior of the WSF sandwich composite plate by testing the cellular size and fabric layer. Based on the above results, we concluded that the compression resistance was excellent, with a higher CV 65% (compression stress value when strain reaches 65%) (Figure 8), which demonstrates the high allowable safe stress and cushion energy absorption of the composite.

### 4.3. Impact Property

A low-speed impact was adopted to explore the performance of the WSF sandwich composite plate. Several aspects of the core pile structure–WSF sandwich composite plate, such as the aspect of core pile density, additional aluminum plate, core material, and impact position, were explored. For core pile density, the WSF sandwich composite plate of carrying capacity increases with the core pile density. Hamid et al. [22] showed the impact: the impact property of the WSF sandwich composite plate layered phenomenon during the impact was strengthened by addition of natural zeolite in core material. Our research group tested the effect of impact position on the core pile and non-core pile properties. In this study, we found that the core pile position had a larger depression diameter during impact, and the lower ground plate was significantly damaged due to the impact energy passed down along the core pile. In comparison, for the non-core pile impact position, the damage was concentrated on the upper ground plate and small damage. Notably, the upper ground plate completely was penetrated when the impact energy was increased (Figure 9a) [24]. The addition of the aluminum plate caused delamination due to the increased impact energy, and the energy was absorbed mainly by crushing the vertical fibers and the supporting foam beneath the region of impact for the WSF sandwich composite plate [13], as shown in Figure 9b.

To date, cross-linking structure–WSF sandwich composite plates have not been investigated. Most studies have focused on the impact properties of the cross-linking structure–hollow WSF sandwich composite plate. Mountasir et al. [29] tested the effect of ground structure, warp, and weft density. They concluded that the impact property of the crimp structure was superior to the non-crimp structure, and it had better impact behavior. In addition, the impact strength and energy absorption increased with the warp and weft density.

### 4.4. Shear and Peel Property

(1)Core pile structure–WSF sandwich composite plate

In the shear test, the stress increased linearly with the strain. Initially, the cracks propagated along the interface between core material and ground plate and stopped when they reached the core piles. The cracks increased tension on the piles, leading to fracture [25]. The study concluded that the shear and peel property resistance increased with ground density because more densely interleaving points improved adhesion between the core material and ground plate [21].

(2)Cross-linking structure–WSF sandwich composite plates

Currently, one study has investigated the cross-linking structure–WSF sandwich composite plates. Therefore, this section explores the effect of different junction methods, such as the cross-linking structure, stitched structure, and adhesive structure, on performance. The failure mode of the cross-linking structure was yarn breakdown, but stitched and adhesive structures underwent delamination in the test. The cross-linking structure’s strength increased by 16% and 39% thanthe stitched structures for U and + types [2] (Figure 10a). Compared with the T-type adhesive structures, the cross-linking structures showed higher stress and 3-fold higher peel force [14] (Figure 10a). Consequently, we concluded that the junction strength increased for the cross-linking structure, followed by the stitched structure and adhesive structure, in that order.

## 5. Conclusions

WSF was reinforced as sandwich composite panels, and this review provides an overview of the performance effects of general fiber material on the sandwich composites plates. The WSF structure on the composite plate was summarized and analyzed. Finally, the processing technology and mechanical properties of the WSF sandwich composite plate were discussed. We gave a summary of the bending, compression, impact, shear, and peel performance of the WSF sandwich composite panels and analyzed their failure mode and failure mechanism. The following key findings were obtained:(1)As reinforcement materials, glass fiber and carbon fiber with high strength and a high modulus have obvious brittle damage during loading. Although polyethylene and polyester has excellent toughness, superior integrity, but they both have a relatively lower carrying capacity.(2)A simple ground-fabrics structure improves the integrity properties of the plate, because of its good stability and symmetry.(3)In the analysis of the structure, the overall performance of the rectangle is better than that of the triangle and trapezoidal structure. However, the transverse compression resistance is small compared with that of the “8” core yarn structure.(4)The addition of nanoparticles or fibers into resin or core materials improves the mechanical properties. Moreover, additional panels can also strengthen the carrying capacity of the plate.(5)The filling and molding process is commonly conducted by using the hand lay-up technology. However, this technology cannot provide uniform and stable plates due to its lower automation and industrialization.(6)The mechanical properties of the weft-WSF sandwich composite plate are better compared with those of the wrap. Moreover, an increase in the internal connection structure density and panel thickness and the reduction in height improve the mechanical properties of the plate.

## Figures and Tables

**Figure 1 polymers-14-03537-f001:**
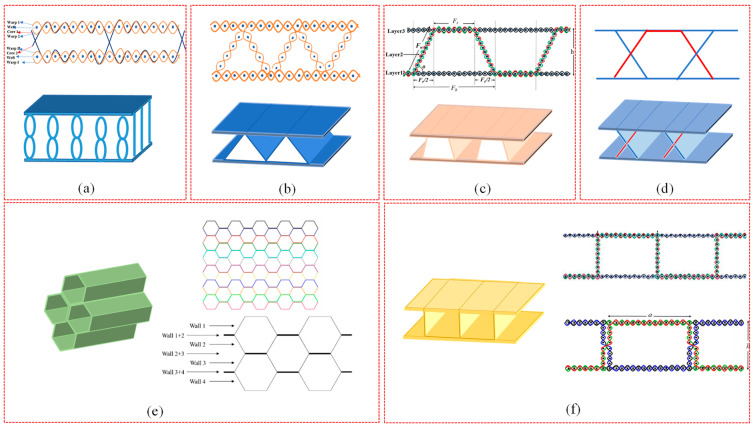
Weaving route and model of WSF: (**a**) “8” core yarn spacer structure, (**b**) triangular (sawtooth) spacer structure, and (**c**) trapezoidal spacer structure (reproduced with permission from Reference [15], Copyright 2019 Taylor & Francis); (**d**) x-spacer structure and (**e**) hexagonal spacer structure (reproduced with permission from Reference [11], Copyright 2019 Elsevier); and (**f**) rectangular spacer structure (reproduced with permission from References [15,33], Copyright 2019 Taylor & Francis, Copyright 2019 Elsevier).

**Figure 2 polymers-14-03537-f002:**
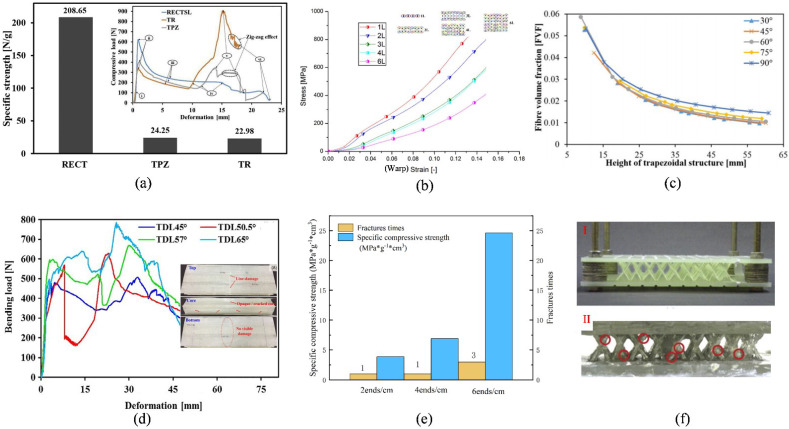
Structure performance comparison: (**a**) comparison of the cross-linking structures (reproduced with permission from Reference [33], Copyright 2019 Elsevier); (**b**) the influence of layers for stress–strain (reproduced with permission from Reference [11], Copyright 2019 Elsevier; (**c**) fiber volume fraction with angle and height (reproduced with permission from Reference [15], Copyright 2019 Taylor & Francis; (**d**) bending load–displacement for different structures (reproduced with permission from Reference [6], Copyright 2020 John Wiley and Sons; (**e**) stress–strain curve of the core pile structure and (**f**) truss woven structure and sandwich failure (reproduced with permission from Reference [41], Copyright 2015 Elsevier).

**Figure 3 polymers-14-03537-f003:**
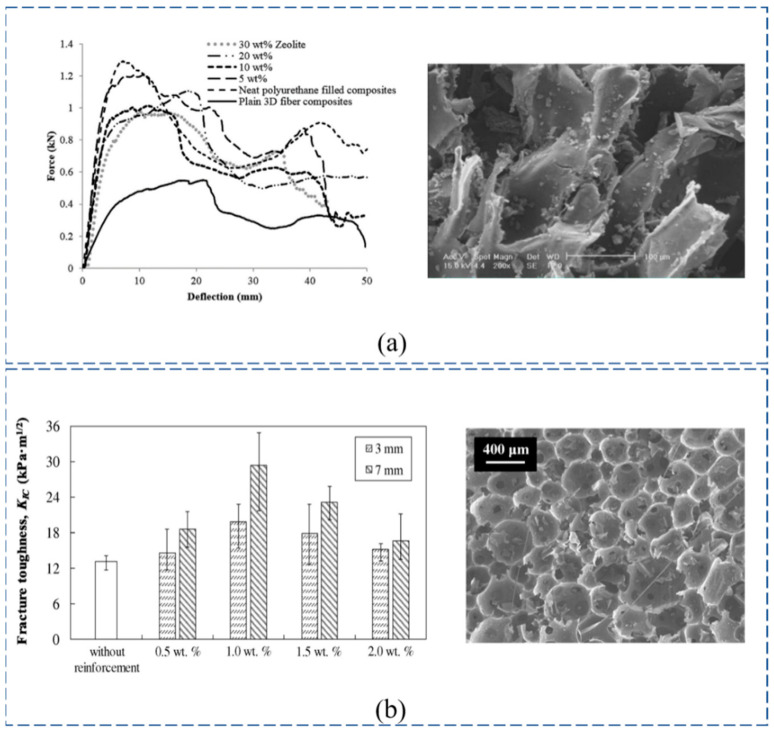
Data diagram and micrograph of the core: (**a**) nanoparticle-enhanced PU foam of loading–displacement and SEM (reproduced with permission from Reference [22], Copyright 2018 Elsevier); (**b**) fiber content increased phenolic foam and SEM (reproduced with permission from Reference [50], Copyright 2016 Elsevier).

**Figure 4 polymers-14-03537-f004:**
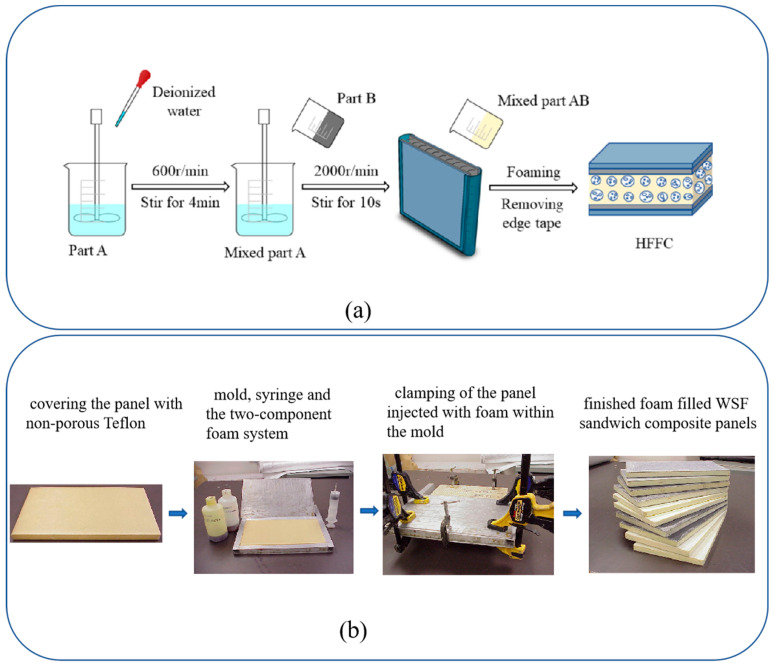
The filling process of WSF sandwich composite plate has two methods: (**a**) does not require the use of molds and (**b**) does require the use of molds (reproduced with permission from References [26,51], Copyright 2007 Elsevier and Copyright 2021 Elsevier).

**Figure 5 polymers-14-03537-f005:**
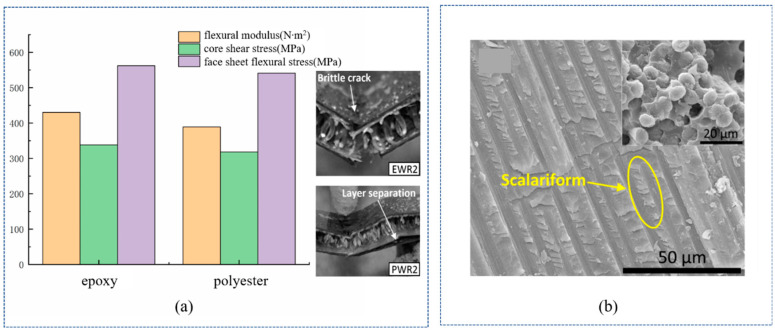
Data and test diagram of the resin matrix: (**a**) performance and failure mode of epoxy resin and polyester resin (reproduced with permission from Reference [60], Copyright 2020 Elsevier); (**b**) SEM images of nano-SiO_2_-reinforced resin (reproduced with permission from Reference [65], Copyright 2020 Elsevier).

**Figure 6 polymers-14-03537-f006:**
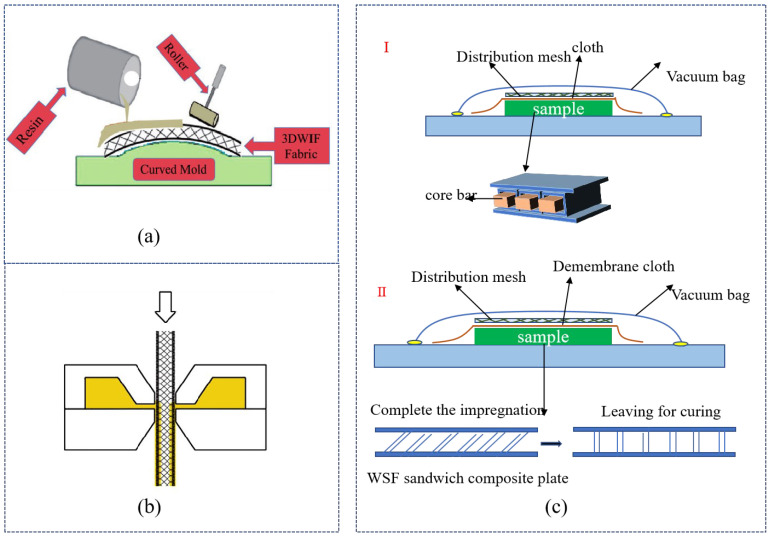
The molding process: (**a**) hand lay-up process (reproduced with permission from Reference [23], Copyright 2018 SAGE Publications); (**b**) slot-coating technology (reproduced with permission from Reference [66], Copyright 2010 Elsevier); (**c**) VIP.

**Figure 7 polymers-14-03537-f007:**
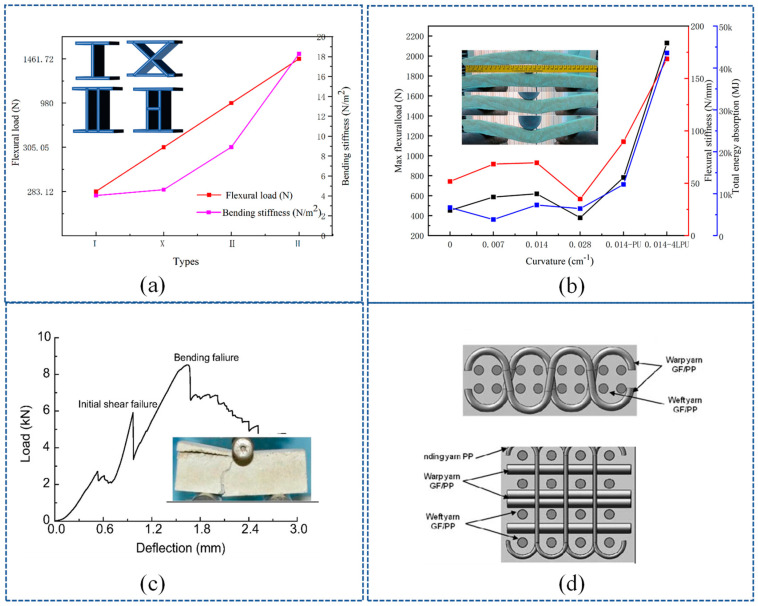
Bending-property test: (**a**) load-displacement of four types and (**b**) curvature the effect of bending property (reproduced with permission from Reference [23], Copyright 2018 SAGE Publications); (**c**) load-displacement and experimental diagram showing the industrial filling material (reproduced with permission from Reference [9], Copyright 2016 Elsevier); and (**d**) non-crimped and crimped structure (reproduced with permission from Reference [29], Copyright 2011 SAGE Publications).

**Figure 8 polymers-14-03537-f008:**
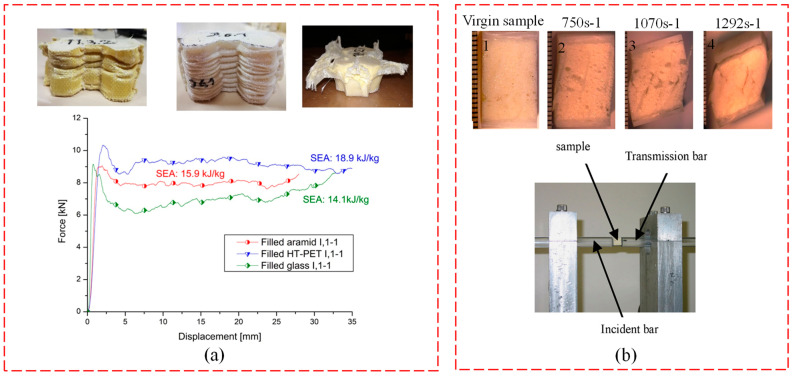
Compression mechanical properties test of WSF sandwich composite plate: (**a**) edgewise compression of the load–displacement curve and experimental diagram of different fibers (reproduced with permission from Reference [11], Copyright 2019 Elsevier); (**b**)dynamic compression test (reproduced with permission from Reference [26], Copyright 2007 Elsevier).

**Figure 9 polymers-14-03537-f009:**
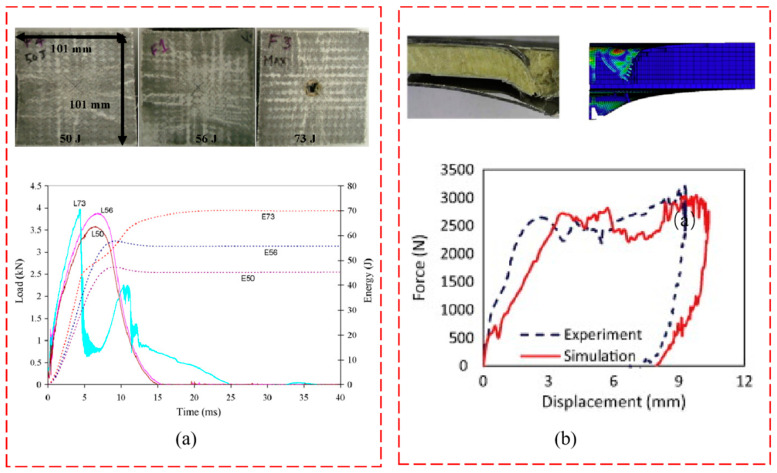
Impact property test of WSF sandwich composite plate. (**a**) the impact energy load–time curve and test damage diagram (reproduced with permission from Reference [13], Copyright 2015 Elsevier). (**b**) Load–displacement curve and failure diagram (reproduced with permission from Reference [24], Copyright 2008 Elsevier).

**Figure 10 polymers-14-03537-f010:**
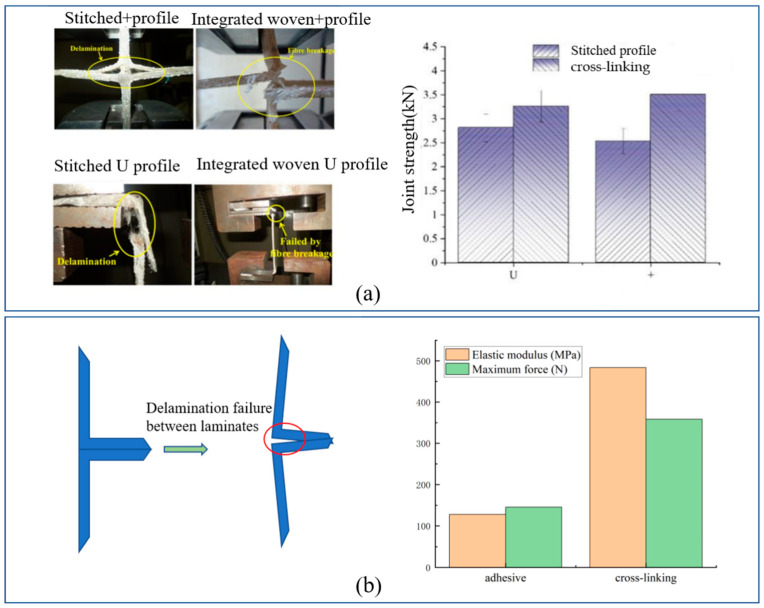
Junction way property test. (**a**) The + and U structure types of the stitched and cross-linking structure of failure mode and property test (reproduced with permission from Reference [2], Copyright 2021 MDPI AG); (**b**) T-type adhesive structure and cross-linking structure of failure mode and property test.

**Table 1 polymers-14-03537-t001:** The studies of WSF sandwich composite plate.

Author	Research Institute	Reinforcement Material	Filling Material	Primary Content	Testing	Reference
Karen De Clerck	Ghent University	Polyester, para-aramid, glass yarns	PU	WSF of structure design	Bending and compression	[11]
Jia Lixia	Hebei University of Science and Technology	Polyester or glass	PU	WSF of structure, weaving, and process technology	Compression, bending, or peel	[17,18,19,20,21]
Hassan Nahvi	Isfahan University of Technology	Glass	PU	Effect of natural nano-structured zeolite	Compression, bending and impact	[22]
Fan Hualin	Nanjing University of Aeronautics and Astronautics	E-glass	foam cementitious	novel ductile cementitious sandwich composites	Compression and bending	[9]
Farid Taheri	Dalhousie University	Glass	PU	add magnesium alloy sheets	Impact	[13]
Sayyed MAHDI Hejazi	Isfahan University of Technology	E-glass	PU	curved	Bending testing	[23]
U.K. Vaidya	University of Alabama at Birmingham	Glass	PU	Effect of impact energy	Impact	[24]
Shaokai Wang	Beihang University	E-glass	PU	WSF of structure design	Shear and compressive	[25]
M.V. Hosur	Tuskegee University	Glass fabric	PU	pressure strain rates	Compressive	[26]

## Data Availability

Not applicable.

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
