# Peer review of "Recent Advances in Woven Spacer Fabric Sandwich Composite Panels: A Review"

_polymers, 2022, doi:10.3390/polym14173537_

Round 1
Reviewer 1 Report
The polymers-1861789 manuscript is a review article related to the technology of WSF composite sandwich panels. It is a specialized article that includes many appropriate references to know the most relevant aspects of this topic.
A review of the topic and the names of the authors was carried out and no documents were found that indicate any type of plagiarism. Its publication without corrections is recommended, although the English language must be reviewed by experts in the field.
Reviewer 2 Report
The data provided here for the review article entitled "The recent advance of woven spacer fabric sandwich composite panels: A review" are interesting; however, I am offering some comments throughout the manuscript including novelty assurance, scientific clarity and tactfulness, and a lot of typo and spacing errors.
(1) The authors did not follow the journal template. Please follow the journal template.
(2) It is needed to use a comma between two or more affiliation indicators (for example, Liu12 should be Liu1, 2).
(3) The authors have mentioned in the abstract that “Finally, the mechanical properties of WSF composite sandwich panels are tested, such as bending, compression,…..” Unfortunately, this is a review article.
(4) The readers would like to grab all of your key findings after having a look at the abstract. But the recent abstract does not convey any idea of the work at all; it looks like a combination of some statements. The abstract must be improved.
(5) Words in the title are not usually needed in the keywords. Remove the same words from the keywords which are mentioned in the title.
(6) The introduction part is unprofessional; there is a lack of consistency between lines and paragraphs, it really needs to be revised very carefully. Some quantitative data should be added to the introduction section.
(7) The main gap of the work is totally absent in the Introduction. The last paragraph of the Introduction should provide information (only) about the science gap in the previous studies and what motivates you to do this review with the objective of the study.
(8) The authors have mentioned in the Material section that “Zhong zhili et al. studied the core spun yarn produced by polypropylene fiber and basalt filament, and found that the breaking strength of wrapped yarn gradually increased with the increase of twist in a certain range” but no proper reference is cited.
(9) Zhang et al. [3] is enough to indicate the proper quotation instead of Zhang Man et al. [3]. This should be maintained for all other citation purposes such as Ghanshyam Neje et al. [2], Hamid Safari et al. [45], Wang He et al. [43], Wang Peng et al. [5], A.S. Vaidya et al. [50], Ferhat Yıldırım et al. [17], Ghanshyam Neje et al. [27], Wang Liyong et al. [4], Tohid Dastan et al. [37], Hamid Safari et al.[42], R.N. Manjunath et al. [10], M.V. Hosur et al. [48], Ruben Geerinck et al. [7], Chunhong Zhu et al. [68].
(10) The authors need to add a new table to compare the results published in the articles.
(11) Please add the copyright/permission to the figure captions if you did not make them yourselves.
(12) The resolution of the figures must be improved.
(13) The texts in Figures 1(a), 1(e), 5(a), 6(c), 10(a) are not sharp.
(14) It is recommended to use “Figure” instead of “Fig.” in the text throughout the manuscript.
(15) Challenges must be included before the conclusion part.
(16) There are lots of typos and grammatical errors observed throughout the manuscript. These must be corrected and revised.
(17) Many related articles are not cited.
(18) References [42] and [45] are the same.
(19) The entire conclusion must be re-written with conclusive findings and by retaining coherence.
(20) References should be according to the journal template.

Round 2
Reviewer 2 Report
The authors have improved the article accordingly. I have no other observations.